

**Title: Vegetation photosynthetic phenology metrics in northern terrestrial ecosystems: a dataset derived from a gross primary productivity product based on solar-induced chlorophyll fluorescence**

Jing Fang[1,2], Xing Li[3], Jingfeng Xiao[4], Xiaodong Yan[5], Bolun Li[3], Feng Liu[1,2]*

CAS Key Laboratory of Aquatic Botany and Watershed Ecology, Wuhan Botanical Garden, Chinese Academy of Sciences, Wuhan 430074, China
Center of Plant Ecology, Core Botanical Gardens, Chinese Academy of Sciences, Wuhan, 430074, China
Research Institute of Agriculture and Life Sciences, Seoul National University, Seoul, South Korea
Earth Systems Research Center, Institute for the Study of Earth, Oceans, and Space, University of New Hampshire, Durham, NH, USA
State Key Laboratory of Earth Surface Processes and Resource Ecology, Faculty of Geographical Science, Beijing Normal University, Beijing 100875, China

Corresponding author: Feng Liu (liufeng@wbgcas.cn)

Postal address of the corresponding author: Chinese Academy of Sciences, Wuhan
430074, China

**Abstract**
Vegetation phenology can profoundly modulate the climate-biosphere interactions and
thus plays a key role in regulating the terrestrial carbon cycle and the climate. However,
most previous phenology studies are based on the traditional vegetation indices, which
are inadequate to characterize the seasonal activity of photosynthesis. Here, we
generated an annual vegetation photosynthetic phenology dataset with a spatial
resolution of 0.05 degree from 2001 to 2020, using the latest gross primary productivity
product based on solar-induced chlorophyll fluorescence (GOSIF-GPP). We combined
smoothing splines with multiple change-point detection to retrieve the phenology
metrics: start of the growing season (SOS), end of the growing season (EOS), and
length of growing season (LOS) for terrestrial ecosystems in the Northern Hemisphere.
We found that the derived phenology metrics agreed better with in situ observations



from the flux tower sites than vegetation indices and MODIS-GPP. Our phenology
metrics captured the spatial-temporal patterns of the single and double growing season
in the Northern Hemisphere. The double season was mainly from the cropland rotation
and ecosystems having two different phenological cycles. In addition, we observed a
trend toward advanced SOS in about 62.98% of the land area, with a mean rate of
$0.14\pm0.01$ days year$^{-1}$, a trend toward delayed EOS in about 61.87% of the area, with a
mean rate of $0.19\pm0.16$ days year$^{-1}$, and a trend toward extended LOS in about 70.52%
of the area, with a mean rate of $0.33\pm0.17$ days year$^{-1}$. Our phenology product can be
used for validating and developing phenology models or carbon cycle models, for
evaluating satellite remote sensing phenology, and for monitoring climate change
impacts    on    terrestrial    ecosystems.    The    data    are    available
at https://doi.org/10.6084/m9.figshare.17195009.v2 (Fang et al. 2021).
**1.  Introduction**
Vegetation phenology, the cycle sequence of plant vital activities, is a highly sensitive
indicator of the climate impacts on terrestrial ecosystems (Richardson et al. 2013, Piao



et al. 2019, Wang et al. 2019, Keenan et al. 2020). Most phenology studies focus on the
structural changes of plants, such as using the growth process of leaf represented by the
greenness indicators (Seyednasrollah et al. 2021, Yang and Noormets 2021). However,
recent studies found that the methods based on vegetation greenness have limited ability
to capture the photosynthesis changes in some vegetation types (e.g. evergreen forests)
since the greenness and photosynthesis are sometimes decoupled (Walther et al. 2016,
Smith et al. 2018). The inaccurate estimation of phenology can lead to substantial
uncertainties in the estimation of plant productivity and carbon sequestration
(Richardson et al. 2012, Wu et al. 2017, Fang et al. 2020).
The plant photosynthetic cycle on the seasonal time scale is termed as 'vegetation
photosynthetic phenology', which represents the functional aspects of plant activities
(Gu et al. 2009). This phenology definition is based on the photosynthesis transition
dates extracted from the gross primary productivity (GPP) time series. Thus, the
accuracy of extracted phenology metrics is largely dependent on the data source of GPP.
Currently, the GPP can either be derived from Eddy Covariance (EC) flux towers at the
ecosystem scale or from satellite remote sensing or modeling at the regional or global



scale (Xiao et al. 2019). The EC technique, which is considered as the most accurate
observation method (Baldocchi et al. 2001), has provided long-term GPP estimates for
more than 20 years. However, these observations are limited by their spatial distribution
and some key areas are still underrepresented (Xiao et al. 2019). For example, only a
few EC sites provide public datasets in the tropical and high latitude regions. GPP
derived from satellite remote sensing is able to investigate large–scale phenology across
the globe (Sjöström et al. 2013). Greenness-related vegetation indices such as the
normalized difference vegetation index (NDVI) and the enhanced vegetation index
(EVI) have been widely used to estimate GPP (Wu et al. 2017, Huang et al. 2019, Dai
et al. 2021). However, these indices work better for capturing the variations in
chlorophyll content or vegetation coverage and are not sufficient to track the
instantaneous physiological changes in vegetation photosynthesis, especially for
evergreen vegetation (Joiner et al. 2014, Li and Xiao 2020). Recently, the emergence
of satellite-based solar-induced chlorophyll fluorescence (SIF) has offered
unprecedented opportunities for developing more accurate photosynthetic phenology
data products on large scales (Joiner et al. 2011, Frankenberg et al. 2014, Li et al. 2018,





Köhler et al. 2018). SIF, a signal emitted by plant chlorophyll molecules after absorbing
photosynthetically active radiation (APAR), is considered to be an effective tool for
diagnosing terrestrial photosynthesis and estimating GPP more accurately (Meroni et
al. 2009, Verma et al. 2017, Wood et al. 2017, Li and Xiao 2020). Based on the SIF
product, recent studies used the relationship between the GPP and SIF to estimate the
regional or global GPP (SIF-GPP) (Li and Xiao 2019, Zhang et al. 2020). Previous
studies reported that SIF-GPP can better capture the GPP dynamics in evergreen
vegetation and dryland ecosystems than traditional vegetation indices (Bertani et al.
2017, Smith et al. 2018).

In addition, the retrieval of phenology in previous studies mainly used a logistic

regression model to fit the time series of smoothed vegetation indices or GPP, and the
predetermined thresholds or inflection points are identified as the transition dates of
phenology in the fitted curve (Garrity et al. 2011, Wang et al. 2017, Yang and Noormets
2021). However, this method needs to reconstruct the original data sequence and thus
results in uncertainty from the model parameterization (Klosterman et al. 2014).
Furthermore, this method is usually used to capture a single growing season instead of



the multiple growing seasons in a given year (Yang and Noormets 2021).
Correspondingly, Richardson et al. (2018) proposed a method of smoothing spline and
multiple change-point detection to retrieve the transition dates of phenology from the
camera data. The strength of this method is not limited by the uncertainty of additional
model parameters and can also be applied in ecosystems having multiple growing
seasons. The method has been successfully used at multiple sites in North America
(Richardson et al. 2018) and needs to be extended to large scales.
Here, we aim to generate a photosynthetic phenology metrics dataset based on the
GPP product derived from satellite SIF data. Our data can detect multiple growing
seasons, which can be used to evaluate the photosynthesis activity of vegetation from
large scales. The metrics include the start state-transition dates of photosynthesis (SOS),
the end state-transition dates of photosynthesis (EOS), and the duration length of
photosynthesis (LOS). With this goal, we constructed a method combining smoothing
filter and change-point detection to retrieve photosynthetic phenology from a recently
developed SIF-based GPP product (GOSIF-GPP: 2001-2020) with a fine spatial
resolution (0.05°). This method enables us to acquire multiple photosynthesis activity



periods of vegetation within one year. The remainder of this paper describes the data of
SIF-GPP and land cover data, the adopted method for retrieving photosynthetic
phenology metrics, the results and discussion of the metrics and their uncertainties, and
the conclusions.


**2. Data**
We used the GOSIF-GPP dataset from 2001-2020 (Li and Xiao 2019) to derive the
phenology metrics on large scales in this study (http://data.globalecology.unh.edu/).
GOSIF-GPP was estimated from the GOSIF dataset based on eight linear SIF-GPP
relationships with 0.05º spatial and 8-day temporal resolutions (i.e., 46 GPP estimates
per year for each 0.05º grid cell). The GOSIF dataset was developed by using discrete
SIF soundings from the Orbiting Carbon Observatory-2 (OCO-2), remote sensing data
from MODIS, and reanalysis data from MERRA-2 based on machine learning method
(Li and Xiao 2019b). The GOSIF-GPP showed reasonable seasonal and spatial patterns
and was highly correlated with GPP from FLUXNET (Li and Xiao 2019). Here, we



identified the vegetation type of each grid cell according to the MODIS Land Cover
Type Product Version 6 (MCD12C1: https://lpdaac.usgs.gov/products/mcd12q1v006/)
(**Fig. S1**, 0.05° spatial resolution). The current study used six broad vegetation types
(i.e. **forests**: evergreen needleleaf forests, evergreen broadleaf forests, deciduous
needleleaf forests, deciduous broadleaf forests, and mixed forests; **shrublands**: closed
canopy shrublands and open shrublands; **savannas**: savannas and woody savannas;
**grasslands**; **wetlands**; **croplands**) in the Northern Hemisphere. For the sake of
reducing noise generated by non-vegetation signals, we excluded the area covered with
bare soil and sparse vegetation (i.e., maximum GPP lower than 2.0 g C m$^{-2}$ day$^{-1}$) (Liu
et al. 2016). Since the seasonal variation of vegetation photosynthesis in the tropical
region is relatively small (Piao et al. 2019), we focused on the area above 30° N latitude.
The final dataset is provided at each 0.05° grid for 20 years in the six terrestrial
ecosystems of the Northern Hemisphere.

To evaluate phenology estimates based on GOSIF-GPP, we used the daily GPP

data from EC flux towers across the Northern Hemisphere based on the
FLUXNET2015 Dataset (https://fluxnet.org/data/fluxnet2015-dataset/) (Pastorello et al.



2020). We retained the EC flux sites that were relatively homogeneous because the
footprint of 0.05º GOSIF product and EC tower may not exactly match (Li and Xiao
2019). We selected the flux sites having available GPP data for more than one year.
The selected flux tower GPP dataset includes 49 sites with 389 site-year data (the
detailed information of these flux sites can be found in **Table S1**). As a comparison, we
also compared the performance of GOSIF-GPP based phenology metrics with those
based on the vegetation indices and GPP products from the MODIS datasets. For each
site, we extracted and calculated three vegetation indices from the Nadir Bidirectional
Reflectance Distribution Function (BRDF)-Adjusted Reflectance dataset MCD43A4
(produced daily and 500 m resolution) including the NDVI, the EVI, the near-infrared
reflectance of vegetation (NIRv) (Badgley et al. 2017), and the 8-day, 500-m MODIS
GPP data (MOD17A2) (Zhao et al. 2005) from 2001 to 2014.


**3. Method**
**3.1 Photosynthetic phenology metrics**





The phenology metrics in this study include SOS, EOS, and LOS. Unlike the traditional
phenological events from the structural changes of leaf or flower, the photosynthetic
phenology is defined as the start (i.e. SOS) and end (i.e. EOS) state-transition dates of
the photosynthesis cycles. These transition dates are used as the phenology metrics.
One full cycle generally has five distinctive stages, including (1) photosynthesis
dormancy period, a season before the growing season; (2) photosynthesis development
period, a GPP rising stage; (3) photosynthesis peak period, a peak stage of GPP; (4)
photosynthesis recession period, a GPP falling stage; and (5) photosynthesis dormancy
period, the photosynthetically inactive stage after the growing season. Most previous
studies used the sigmoid-based methods (e.g., double-logistic model) to extract the
phenology, but these methods are limited to the single cycle (Yang and Noormets 2021).
Because some regions or ecosystems had multiple cycles in one year, we used the
smoothing splines and change points to identify the transition dates of photosynthesis.
In this study, all transition dates were extracted from the daily GPP sequence of each
grid cell. Thus, we interpolated the 8-day GOSIF-GPP data to the daily scale using
cubic spline interpolation before the extraction.

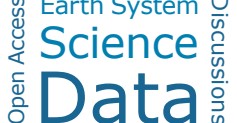

We constructed an automatic method to retrieve transition dates (i.e. SOS and EOS)

of photosynthetic phenology using GPP data. The algorithm of this method is outlined

in the flowchart in **Fig. 1**. The important basis for acquiring phenological events was

the data reconstruction using smoothing methods to minimize the impact of abnormal

values (Li et al. 2019). We applied the iterative procedure to conduct the smoothing

process (**Fig. 1**): (1) Smoothing the GPP time series by the Savitzky-Golay filter, which

can reflect the change characteristics of the original data sequence; (2) Calculating the

ratio of the daily GPP value to the smooth value; (3) Identifying outliers in these ratios

by using the Grubbs test; (4) Using the smooth value instead of the daily GPP value

when the ratios were larger than one standard deviation below the mean ratio; (5)

Applying the iterative procedure up to 20 times or until no outliers were detected from

one iteration to the next. This procedure can largely keep the raw seasonal pattern of

photosynthesis and avoid the uncertainty of parameter estimation by reconstructing the

data time series by estimating parameters in the double logistic model.

The potential change points in the final smoothing splines were identified with the

Pruned Exact Linear Time (PELT) method. This method can accurately detect the



significant change points in the data time series and does not need to preset the number
of change points. The PELT was first applied by Killick et al. (2012), and they described
in detail on how to find the change points in time series. For each photosynthesis cycle,
we followed Richardson et al. (2018) to set the penalty factor and the minimum segment
length of PELT as 0.5 and 14-days, respectively. We calculated the mean GPP value of
the adjacent change points as the potential peak and bottom baseline in one full cycle.
According to the time series of mean GPP value, we used the difference method to
detect the bottoms and peaks (i.e., minimum and maximum value in each cycle). The
adjacent bottoms and one peak were formed as a full cycle, and the value of these points
was considered as the baselines. Some GOSIF-GPP data affected by the weak
vegetation SIF signals could have unreliable cycles, and these cycles that had peaks less
than 0.25 of the maximum peak were excluded in the current study.

Here, the SOS and EOS dates of each cycle were determined by amplitude

thresholds. The amplitude was equal to the peak minus the bottom. Although the "true"
onset of photosynthesis may correspond most closely to the 10% amplitude threshold
(Wu et al. 2017), the most tightly-constrained transition dates tended to occur in the





later dates of the GPP rising stage and the earlier dates of the GPP falling stage
(Richardson et al. 2018). Thus, we followed Richardson et al. (2018) to provide the
SOS and EOS dates by using three amplitude thresholds: 10%, 25%, and 50%. The
SOS and EOS were determined when the GPP smoothing splines reached the value of
amplitude thresholds, and the LOS was defined as EOS minus SOS:

$$SOS_i = t, if\ GPP_S(t) = (Peak - Bottom_1) \times threshold_i \qquad (1)$$

$$EOS_i = t, if\ GPP_S(t) = (Peak - Bottom_2) \times threshold_i \qquad (2)$$

$$LOS_i = EOS_i - SOS_i \qquad (3)$$

where $i$ is the threshold (10%, 25%, and 50%); $t$ is the day of the year (DOY); $GPP_S$ is
the daily value of the smoothing splines; $Bottom_1$ is the baseline for dormancy season
before the growing season; $Bottom_2$ is the baseline for dormancy season after growing
season. Note that we retrieved the phenology of vegetation indices (i.e. daily data) and
MODIS-GPP (i.e. interpolating the 8-day data to the daily scale) by using the same
method.

**3.2 Uncertainty estimation**

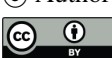



The uncertainties in the estimates of phenology metrics mainly arise from the gridded
SIF-based GPP estimates, such as using the limited explanatory variables to acquire the
gridded SIF estimates (i.e., GOSIF) and the relationship between the SIF and GPP. In
this study, we did not assess the quality of the underlying SIF and GPP data, which was
previously evaluated (Li and Xiao 2019); instead we used the Monte Carlo
Bootstrapping method (Efron 1992) to estimate the related uncertainties. Bootstrapping
provides valuable information about uncertainties without making assumptions about
the underlying data distributions (Elmore et al. 2012). For each year of the individual
grid cell, we used bootstrapping to replace the transition dates with 100 times random
uniform sampling (Yang and Noormets 2021). The 5th and 95th percentiles of the 100
bootstrapped data were considered as the confidence interval of the mean estimated
from the original transition dates.


**4. Results and discussion**
**4.1 Comparison with phenology derived from vegetation indices, MODIS-GPP,**



**and EC tower data**
We used the photosynthetic phenology metrics extracted from the daily GPP of the flux
towers to examine the corresponding metrics extracted from the GOSIF-GPP product.
We also use the same method to retrieve phenology from the NDVI, EVI, NIR$_v$, and
MODIS GPP for the EC tower sites. According to the different thresholds, the metrics
were divided in to nine groups (SOS$_{10\%}$, SOS$_{25\%}$, SOS$_{50\%}$: SOS with 10%, 25%, and
50% amplitude threshold; EOS$_{10\%}$, EOS$_{25\%}$, EOS$_{50\%}$: EOS with 10%, 25%, and 50%
amplitude threshold; EOS$_{10\%}$, EOS$_{25\%}$, EOS$_{50\%}$: EOS with 10%, 25%, and 50%
amplitude threshold) (**Fig. 2** and **Table 1**). Overall, the phenology metrics of GOSIF-
GPP showed the highest correlations with the phenology metrics of EC tower GPP,
while the phenology of NDVI showed the lowest correlations. For each metric, (1) SOS,
the correlation coefficient ($R$) between the 10%, 25%, and 50% SOS of EC tower GPP
and other data were (i.e. from high to low): GOSIF-GPP (0.78-0.80), MODIS-GPP
(0.65-0.67), NIR$_v$ (0.47-0.60), EVI (0.40-0.57), and NDVI (0.14-0.39); the root mean
square error ($RMSE$) were (i.e. from low to high): GOSIF-GPP (14.99-18.03 days),
MODIS-GPP (18.83-22.98 days), NIR$_v$ (21.74-29.86 days), EVI (24.97-36.97 days),



and NDVI (26.20-26.91 days). (2) EOS, the highest $R$ between 10%, 25%, and 50%
EOS of EC tower GPP and other data was GOSIF-GPP (0.63-0.73) and the lowest $R$
was NDVI (0.42-0.56). (3) LOS, the highest $R$ between 10%, 25%, and 50% EOS of
EC tower GPP and other data was GOSIF-GPP (0.65-0.76) and the lowest $R$ was NDVI
(0.28-0.40). The comparisons indicated that GOSIF-GPP showed consistently better
performance than the vegetation indices (i.e., NDVI, EVI, and NIRv) for different
phenology metrics and different thresholds. MODIS-GPP had larger deviations
compared to GOSIF-GPP, which highlights the need for the improvement on light use
efficient models. NIRv, the product of near-infrared reflectance and NDVI (Badgley et
al. 2017), was slightly better to capture the phenology metrics of tower GPP than EVI
and NDVI. The results agreed with previous studies which showed a stronger ability of
SIF in responding to the environmental conditions such as water and heat stresses, and
thus in better capturing the seasonal and interannual photosynthetic activity (Walther
et al. 2016; Smith et al. 2018; Li et al. 2018).

The derived phenology of GOSIF-GPP and EC tower GPP showed a close

correspondence across the 389 site-years. The best performance of the different



thresholds in SOS, EOS, and LOS was 25% (R=0.80, 0.73, and 0.76; RMSE=15.83,
21.89, and 29.14 days, respectively), and the threshold of 10% had relatively low
performance in SOS and EOS (R=0.79 and 0.63; RMSE=18.03 and 23.55 days,
respectively) and 50% had relatively low performance in LOS (R=0.65; RMSE=27.89
days). Our results showed that our method better captured the SOS than the EOS, which
was consistent with previous studies that uncertainty occurred in satellite-based EOS
estimations, especially for the evergreen vegetation such as tropical and boreal
evergreen forests (Liu et al. 2016, Piao et al. 2019). In addition, more tower sites need
to be considered in further studies so that the photosynthesis phenology metrics from
the SIF product can be better evaluated.

**4.2 Number of growing seasons**
We used the method to retrieve the multiple growing seasons in the Northern
Hemisphere. **Fig. 3** showed the spatial distribution of the number of growing seasons.
Most regions in the Northern Hemisphere had a single growing season, while part of
the cropland had a double growing season in a given year. The North China Plain (the



red part in the top right of **Fig. 3**) had the most areas with the double growing season
because the wheat-maize rotation was the most important cropping system in this plain
(Zhao et al. 2006). This artificial crop rotation pattern brought two photosynthesis
cycles: wheat grows in winter and spring, and maize grows in summer and autumn. In
addition to croplands, a small proportion of the grid cells also had double growing
seasons, such as some areas in California. Turner et al. (2020) reported that the double
growing season in California was due to two overlapping ecosystems in one grid,
whereas they were out of phase with each other: grasslands showed a peak of the
growing season in April and forests peak in June. The phenology retrieval of such
mixed ecosystems is still challenging and requires further exploration (Piao et al. 2019).

**4.3 Spatial patterns of photosynthetic phenology metrics**
Here, we showed the spatial distribution of the first growing season in **Fig. 4**. Overall,
phenology metrics (SOS, EOS, and LOS) in terrestrial ecosystems of the Northern
Hemisphere exhibited a spatially explicit pattern from the high latitudes to the low
latitudes. Limited by low temperature, the areas around the Arctic Circle had the latest
SOS (DOY>140), the earliest EOS (DOY<220), and the shortest LOS (days<120). LOS
gradually increased as the climate conditions became more suitable for photosynthesis
and then reached the longest in the subtropics. For different thresholds, the mean
difference of SOS, EOS, and LOS between 10% and 50% was 30 days, 40 days, and
70 days, respectively. For different ecosystems (**Table 2**), grasslands showed the
earliest SOS and EOS among all biomes; forests and savannas had the latest SOS;
croplands and forests exhibited the latest EOS and the longest LOS, while shrublands
and grasslands had the shortest LOS.
**Fig. 5** showed the spatial distribution of the second growing season. We found that
the second 10% SOS in North China Plain was in the end of May (DOY=150) and the
second 10% EOS was in the middle of September (DOY=280). This was consistent
with the emergence and dormancy of maize (i.e. the second growing season). The wheat
would seed after the maize was harvested and the greenness of wheat was in the early
March of the next year, which was the start time of the first growing season (Tang et al.
2020). In California, the second 10% SOS of some areas was in the early June
(DOY=160) and 10% EOS was in the middle of September (DOY=280); the second



10% SOS of other areas was in the late September (DOY=250) and 10% EOS was in
the late November (DOY=330). These results were from the two different mixed grids,
one included the evergreen forests and grasslands, another included the croplands and
grasslands. Turner et al. (2020) found that the growth of grasses provides the first
growing season for these grids. As the grids included evergreen forests entered summer,
the increase of the available water in the soil resulted in the growth of evergreen woody
plants, prompting these grids to enter the second growing season. Other ecosystems
were gradually entered the dormant stage in fall, but the crops still maintained
photosynthesis, making the grids containing croplands show the second growing season.

**4.4 Uncertainties of photosynthetic phenology metrics**
The uncertainty used in this study was defined as the 5th and 95th percentiles of the
100 Monte Carlo bootstrapping samples ranging from a few days to several weeks
(**Table 2**). The uncertainty was the lowest for SOS and the highest for LOS; EOS had
intermediate uncertainty. The highest uncertainty in LOS maybe because of the
compounding effect of SOS and EOS (Yang and Noormets 2021). Generally, metrics





of grasslands had the lowest uncertainty: SOS uncertainty ranged from 3.8 to 5.1 days,
EOS uncertainty ranged from 8.6 to 10.4 days, LOS uncertainty ranged from 13.7 to
14.2 days; forests have the largest uncertainty, with SOS uncertainty ranging from 7.3
to 9.4 days, EOS uncertainty from 16.0 to 18.4 days, and LOS uncertainty from 25.4 to
25.7 days. The high uncertainty in the forests was possibly because this ecosystem
included multiple mixed vegetation types and the phenology of these plants was more
difficult to retrieve (Piao et al. 2019).

**322    4.5 Changes in photosynthetic phenology metrics**

We conducted the linear regression analysis by using the transition dates of phenology
and the time series in each grid cell, and the regression coefficient was considered as
the changing trend of the grid cell (**Fig. 6**). Here, we only showed the changes of the
dominant single growing season. For the spatial distribution of the phenology metrics
with the three thresholds, 61.71-64.25% of the study area experienced advanced trends
of SOS, with a large advanced trend in northwestern North America, northern Siberia,
and eastern Europe (changes>0.6 days year$^{-1}$); 57.97%-65.89% of the study area



experienced delayed trends of EOS, with a large delayed trend in the northern North
America and northern Siberia (changes>0.4 days year$^{-1}$); 70.29-70.76% of the study
area experienced extended trends of LOS, with a large extended trend in northern China,
northern North America, and northern Siberia (changes>0.6 days year$^{-1}$). Note that the
inconsistent climate change trends in different seasons may lead to advanced or delayed
SOS and EOS in some regions simultaneously, such as eastern Europe (Cohen et al.

2012).

We spatially averaged the phenology metrics for the terrestrial ecosystems across

the Northern Hemisphere to assess the interannual variation of phenology metrics (**Fig.**
**7**). During the period 2001-2020, the mean SOS of all thresholds significantly advanced
by 0.13-0.16 days year$^{-1}$ ($p<0.05$); the mean EOS of 10% and 25% significantly
advanced by 0.03-0.35 days year$^{-1}$ ($p<0.05$); the mean LOS of all thresholds
significantly extended by 0.16-0.51 days year$^{-1}$ ($p<0.01$). These findings are consistent
with previous studies (Zhu et al. 2012, Liu et al. 2016). For example, Liu et al. (2016)
indicated that the EOS delayed by 0.18 days year$^{-1}$ across the Northern Hemisphere of

1982-2011.







## 5. Data availability

This dataset is divided into single and double growing seasons. The entire dataset is

deposited        at        the        open-access        repository        Figshare

(https://doi.org/10.6084/m9.figshare.17195009.v2; Fang et al. 2021).

## 6. Conclusions

This study used a long-term (2001-2020) SIF-based GPP product (GOSIF-GPP) to

generate annual photosynthetic phenology of vegetation with a high spatial resolution

(0.05º) in the Northern Hemisphere. Here, we applied a method combining filter

smoothing and change point detection to determine the annual dynamics of phenology

metrics (i.e., SOS, EOS, and LOS). This method avoided the re-modeling of the GPP

time series and allowed the extraction of metrics with different thresholds in multiple

growing seasons. We provided data users with three choices (10%, 25%, and 50%





threshold) of the metrics most appropriate for their specific application. Overall, the
photosynthetic phenology metrics based on GOSIF-GPP agree with those extracted
from in situ observations of EC towers. Compared to the metrics of vegetation indices
and MODIS-GPP, the GOSIF-GPP metrics can provided more accurate phenology in
most EC tower sites. The comparison with field data acquired at the EC towers suggests
the 25% threshold of GOSIF-GPP can better capture the dynamics of photosynthetic
phenology than other thresholds. In addition, the results showed a spatially explicit
pattern from the north to the south in Northern Hemisphere. The SOS of all thresholds
presented a significant advanced trend in the past 20 years; the EOS of 50% threshold
showed an insignificant delayed trend; the LOS of all thresholds had a significant
extended trend.

The phenology product based on GOSIF-GPP in our study is of great use in

vegetation phenology studies because the SIF can directly reveal seasonal variations in
vegetation vital activities (Mohammed et al. 2019). With these metrics, the response of
vegetation phenology to climate change can be further investigated such as the
importance of precipitation in spring phenology (Li et al. 2021). It will also be useful



for developing and validating dynamic vegetation models. Our phenology metrics
could be further improved when more accurate SIF-based GPP estimates are available.


**Acknowledgments**
This study was supported by the National Natural Science Foundation of China
(32101349, 32171599). This study also was supported by the National Key R&D
Program of China (2019YFA0606904) and the Key Program of the National Natural
Science Foundation of China (32130069). J.X. was supported by the University of New
Hampshire.

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

**Table 1.** Statistical comparison of the phenology metrics retrieved from EC tower GPP and GOSIF-GPP, NDVI, EVI, NIRv, and MODIS-GPP. 10%, 25%, and 50% mean the three thresholds. The bold means the highest *R* and the lowest *RMSE*. *R*: correlation coefficient; *RMSE*: root mean square error.

| Data source | SOS | | | EOS | | | LOS | | |
|---|---|---|---|---|---|---|---|---|---|
| | 10% | 25% | 50% | 10% | 25% | 50% | 10% | 25% | 50% |
| | | | | | | ***R*** | | | | |
| GOSIF-GPP | **0.79** | **0.80** | **0.78** | **0.63** | **0.73** | 0.63 | **0.72** | **0.76** | **0.65** |
| NDVI | 0.14 | 0.25 | 0.39 | 0.45 | 0.42 | 0.56 | 0.28 | 0.32 | 0.40 |
| EVI | 0.40 | 0.46 | 0.57 | 0.57 | 0.60 | 0.66 | 0.37 | 0.37 | 0.38 |
| NIRv | 0.47 | 0.51 | 0.60 | **0.63** | 0.66 | **0.67** | 0.51 | 0.48 | 0.41 |
| MODIS-GPP | 0.66 | 0.67 | 0.65 | 0.29 | 0.55 | 0.61 | 0.47 | 0.55 | 0.49 |
| | | | | | | ***RMSE* (days)** | | | | |
| GOSIF-GPP | **18.03** | **15.83** | **14.99** | **23.55** | **21.89** | 24.38 | **33.93** | **29.14** | **27.89** |



| | | | | | | | | | |
|---|---|---|---|---|---|---|---|---|---|
| NDVI | 36.91 | 32.18 | 26.20 | 34.13 | 39.68 | 41.86 | 53.87 | 53.92 | 52.39 |
| EVI | 36.97 | 31.34 | 24.97 | 31.75 | 29.41 | 26.09 | 58.53 | 49.04 | 39.28 |
| NIRv | 29.86 | 26.00 | 21.74 | 27.26 | 25.67 | 24.95 | 46.11 | 40.03 | 35.12 |
| MODIS-GPP | 22.98 | 20.56 | 18.83 | 30.22 | 24.76 | **23.88** | 43.79 | 36.33 | 32.25 |

**Table 2.** The mean value and uncertainty of photosynthetic phenology metrics in the different terrestrial ecosystems.

| Terrestrial ecosystems | Threshold | Mean SOS (uncertainty) | Mean EOS (uncertainty) | Mean LOS (uncertainty) |
|---|---|---|---|---|
| Forests | 10% | 108.26 (7.34) | 271.29 (18.39) | 163.03 (25.73) |
| | 25% | 122.12 (8.28) | 255.40 (17.31) | 133.28 (25.59) |
| | 50% | 138.44 (9.38) | 236.24 (16.01) | 97.80 (25.40) |
| Shrublands | 10% | 75.10 (5.09) | 144.33 (9.78) | 69.22 (14.88) |
| | 25% | 80.97 (5.49) | 137.14 (9.30) | 56.17 (14.79) |
| | 50% | 88.14 (5.97) | 129.02 (8.75) | 40.89 (14.72) |
| Savannas | 10% | 106.37 (7.21) | 244.25 (16.56) | 137.88 (23.77) |
| | 25% | 117.25 (7.95) | 230.10 (15.60) | 112.85 (23.55) |
| | 50% | 130.85 (8.87) | 213.10 (14.45) | 82.24 (23.32) |
| Grasslands | 10% | 56.72 (3.84) | 153.48 (10.40) | 96.76 (14.25) |
| | 25% | 65.21 (4.42) | 140.37 (9.52) | 75.16 (13.94) |
| | 50% | 74.71 (5.06) | 127.55 (8.65) | 52.84 (13.71) |
| Wetlands | 10% | 106.58 (7.23) | 213.13 (14.45) | 106.55 (21.67) |
| | 25% | 115.24 (7.81) | 202.70 (13.70) | 86.83 (21.51) |
| | 50% | 126.24 (8.56) | 189.61 (12.85) | 63.37 (21.41) |
| Croplands | 10% | 86.60 (5.87) | 272.47 (18.47) | 185.87 (24.34) |
| | 25% | 102.20 (6.93) | 250.78 (17.00) | 148.58 (23.93) |
| | 50% | 120.56 (8.17) | 226.01 (15.32) | 105.45 (23.49) |



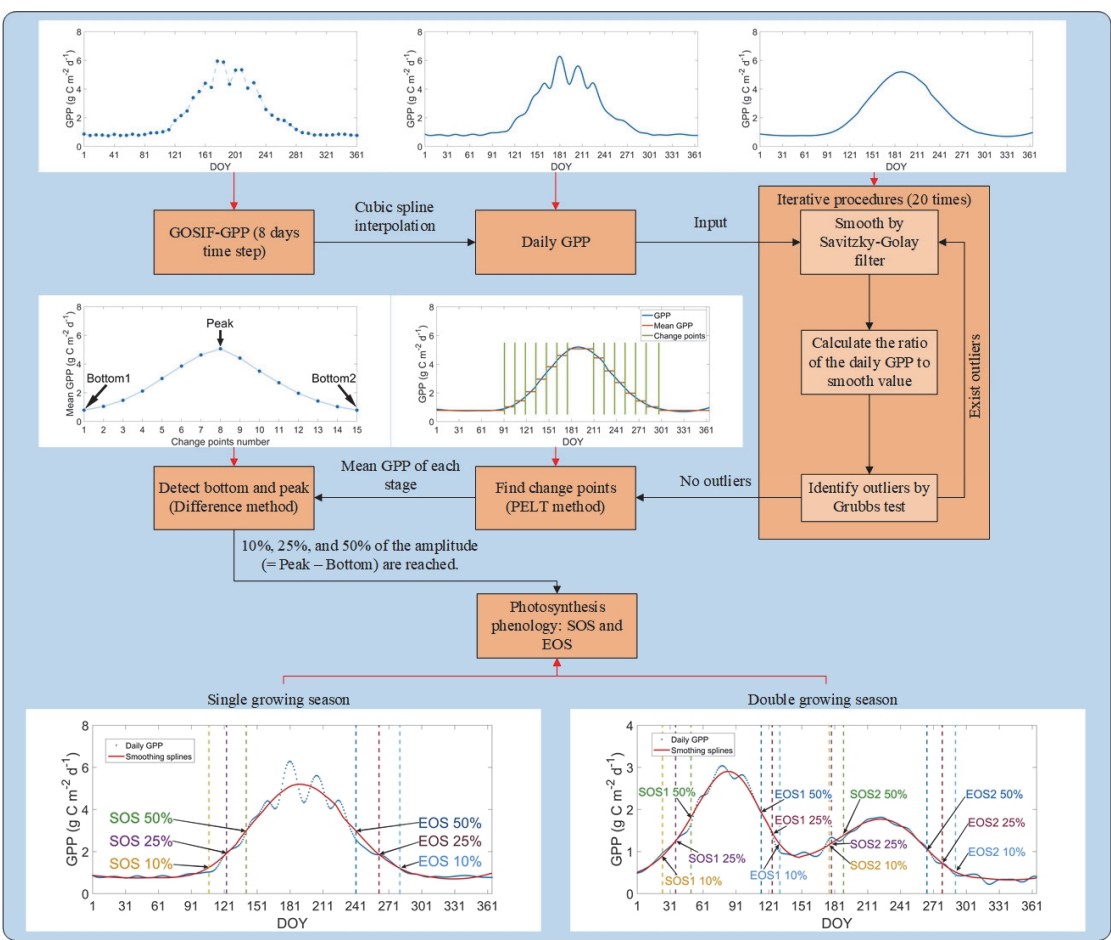

**Fig. 1.** Illustration of the method for identifying the transition dates of photosynthetic phenology. The method is based on three thresholds, 10%, 25%, and 50%. Bottom1: baseline for dormancy season before the growing season; Peak: the peak value in one single cycle; Bottom2: baseline for dormancy season after growing season. The example of the single growing season is from one forest site (latitude: 60.0º N, longitude: 15.5º E); the example of the double growing season is from one cropland site (latitude: 36.5º N, longitude: 36.0º E).

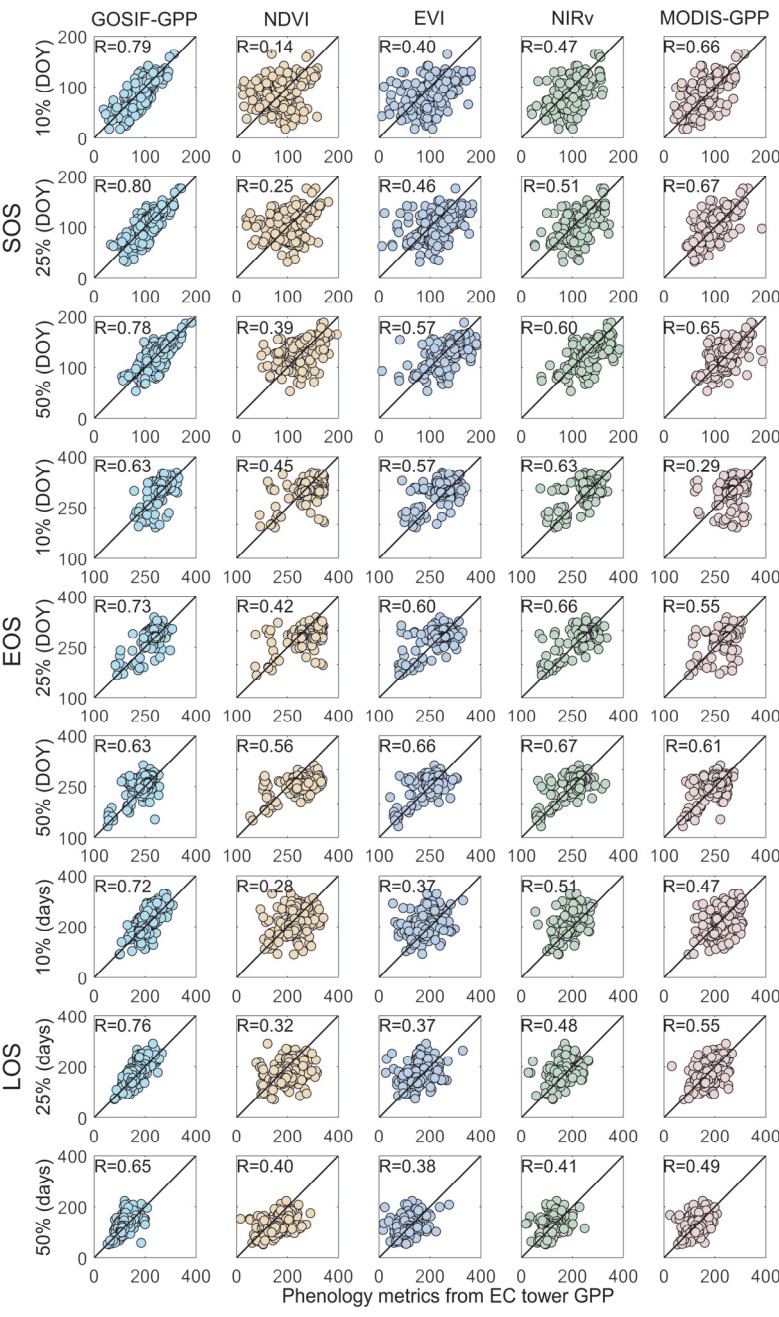

**Fig. 2.** The comparison of the phenology metrics retrieves from EC tower GPP and
GOSIF-GPP, NDVI, EVI, NIRv, and MODIS-GPP. The dotted line represents a 1:1
line. DOY: day of the year; R: correlation coefficient.

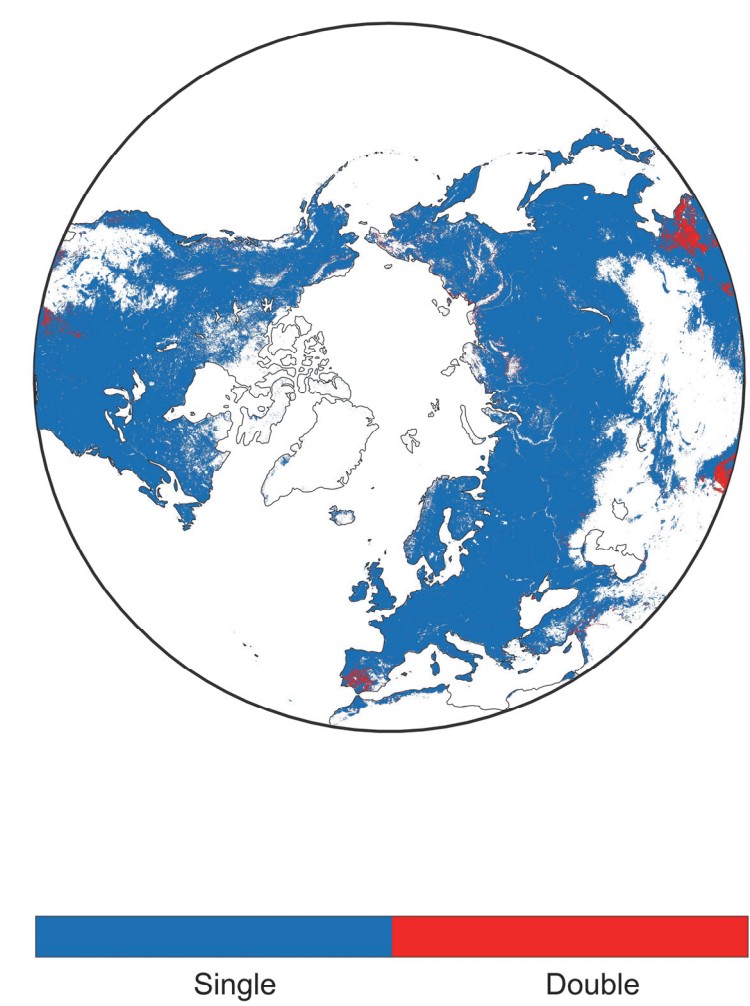

**Fig. 3.** The spatial distribution of the number of growing seasons in the Northern Hemisphere (0.05° spatial resolution).



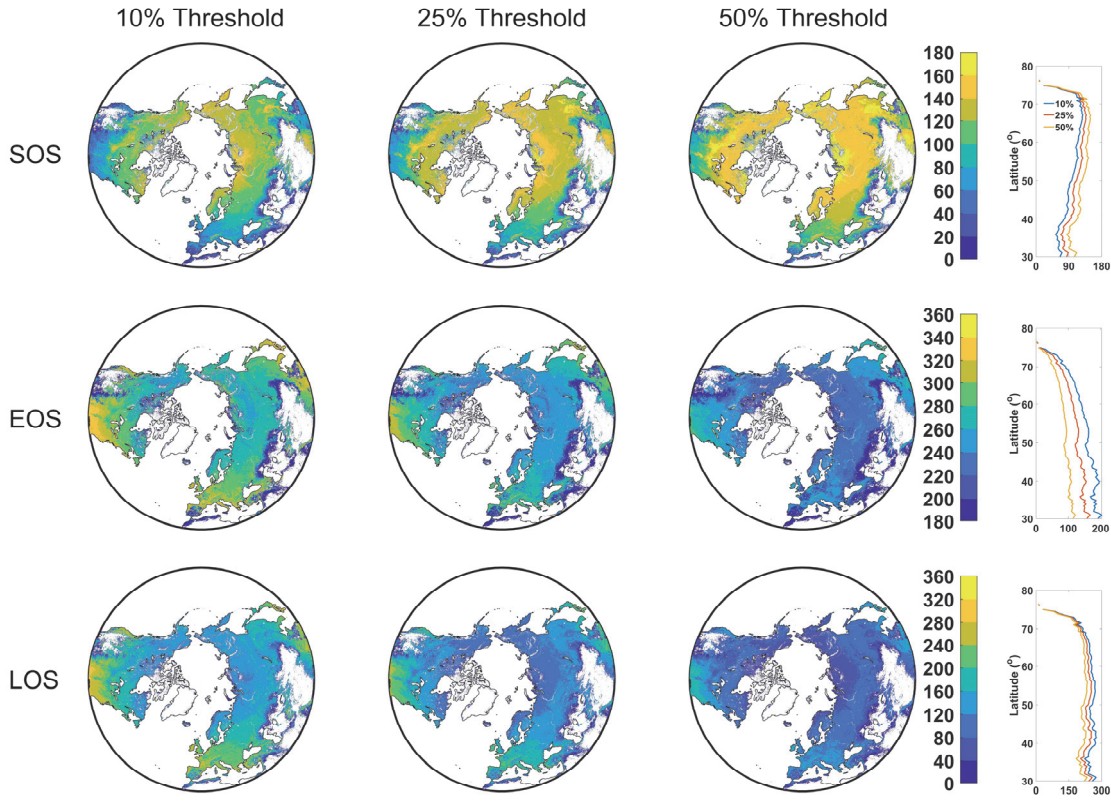

**Fig. 4.** The spatial distribution of the mean photosynthetic phenology metrics (first growing season) in the Northern Hemisphere of 2001-2020 (0.05° spatial resolution). The right parts are the latitudinal pattern.

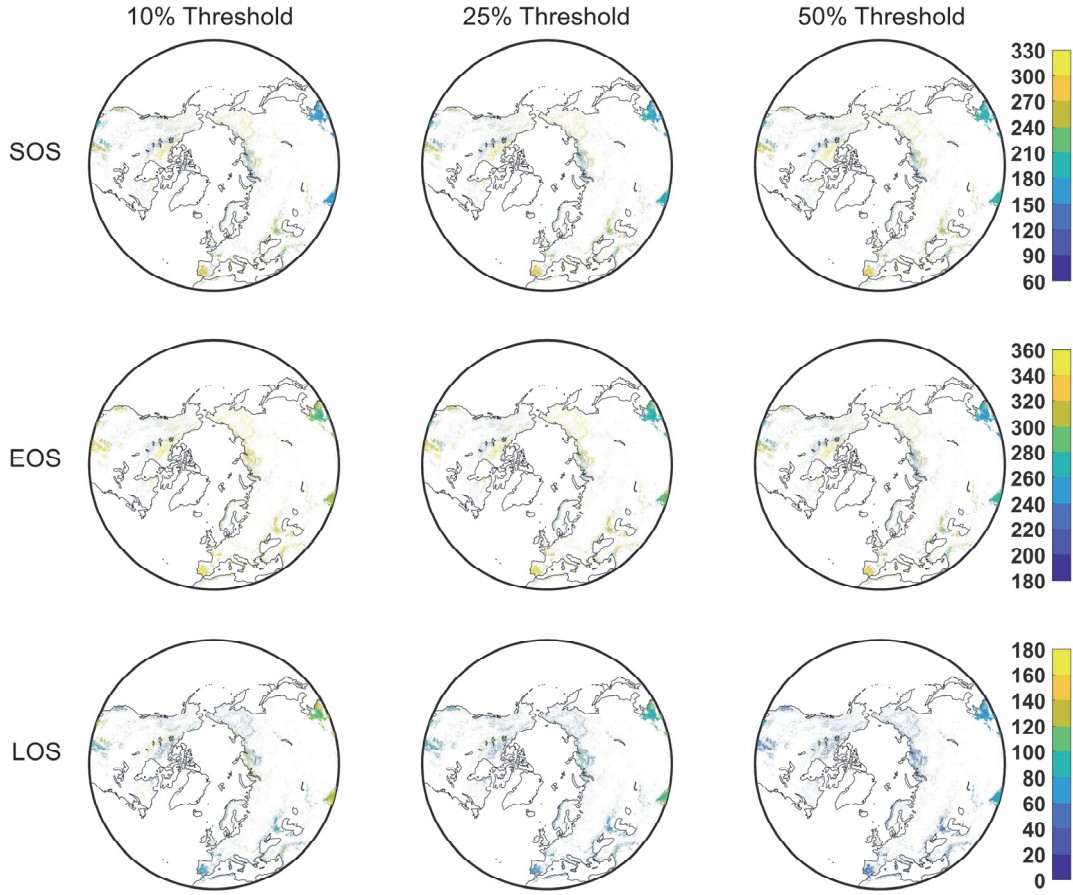

**Fig. 5.** The spatial distribution of the mean photosynthetic phenology metrics of the second growing season in the Northern Hemisphere of 2001-2020 (0.05º spatial resolution).

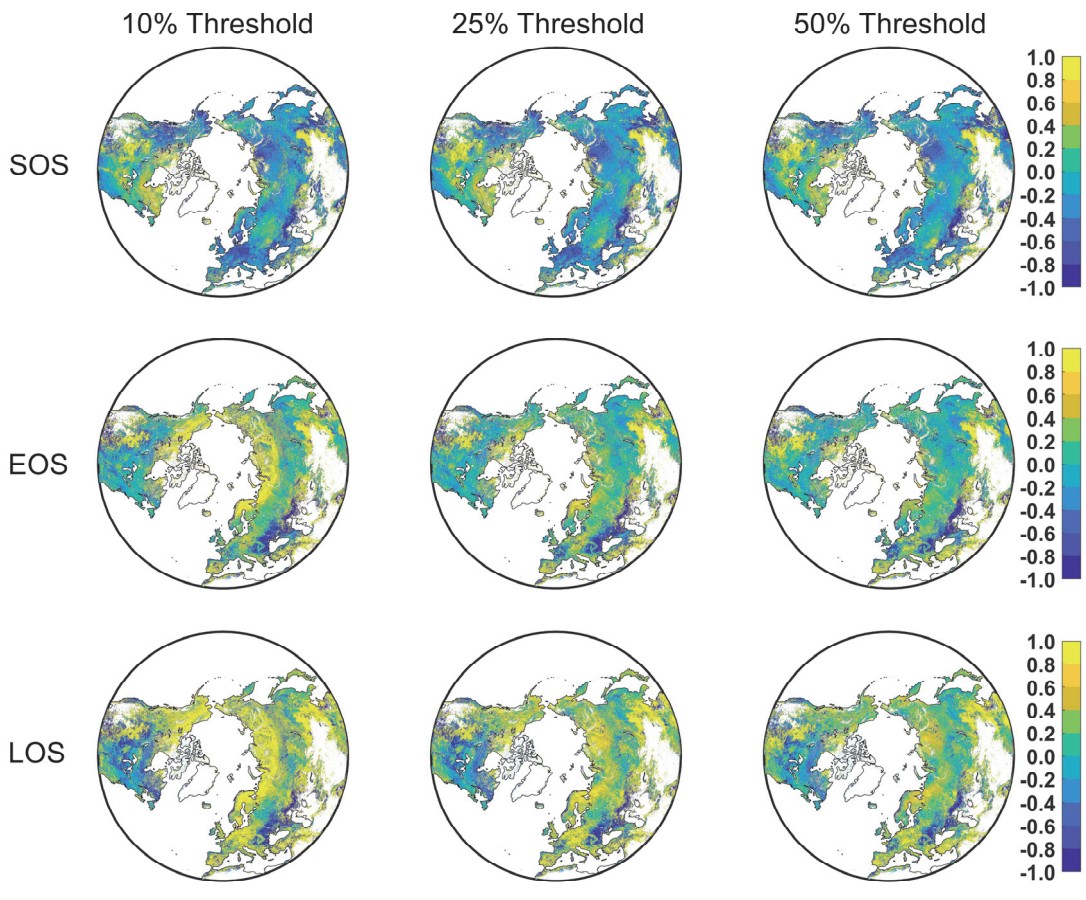

**Fig. 6.** Changes in photosynthetic phenology metrics in the Northern Hemisphere over the period 2001-2020.

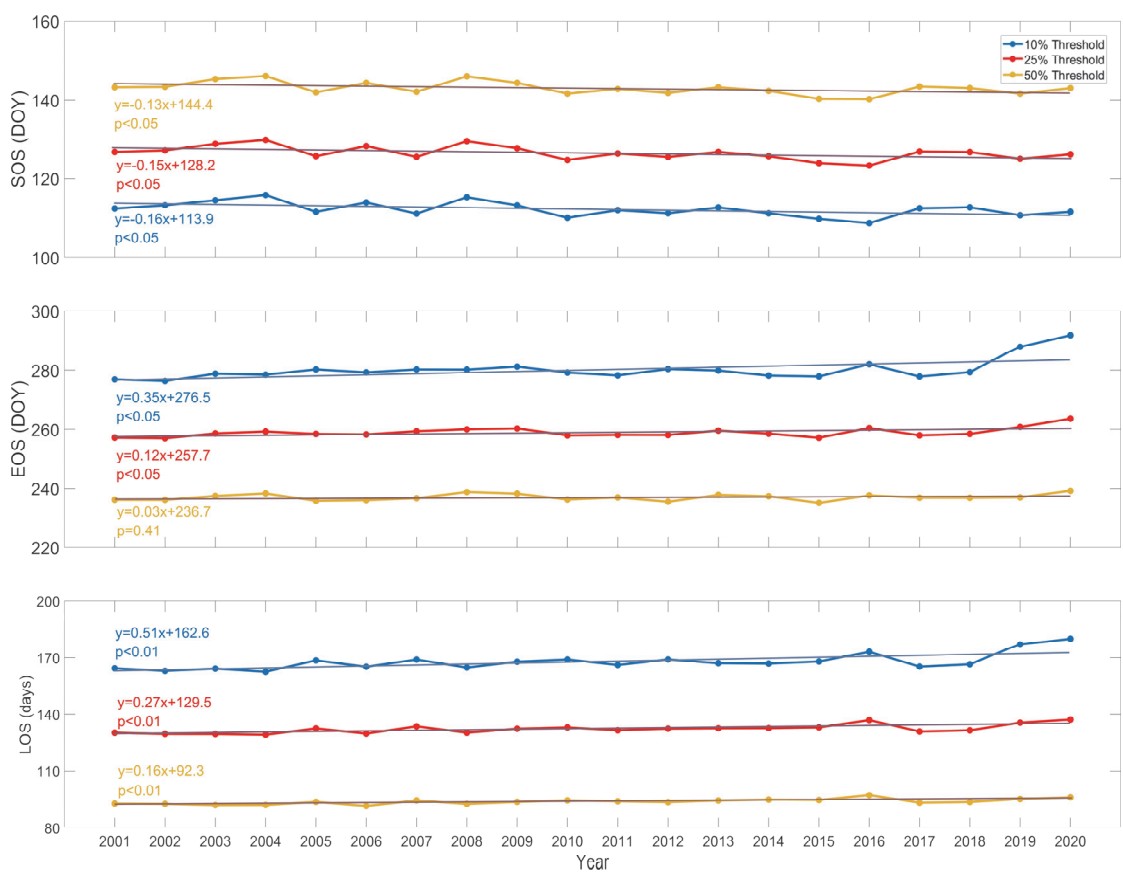

**Fig. 7.** Annual photosynthetic phenology metrics in the Northern Hemisphere during 2001-2020. The straight lines represent the change trends.