# Peer review of "Title: Vegetation photosynthetic phenology metrics in northern terrestrial ecosystems: a dataset derived from a gross primary productivity product based on solar-induced chlorophyll fluorescence"

_Earth System Science Data, 2021_

## Author Comment (AC1)

**RESPONSE TO REVIEWERS' COMMENTS**

Dear Prof. Tian and reviewer,

We would like to express our great appreciation to you for handling and reviewing our manuscript entitled "Vegetation photosynthetic phenology metrics in northern terrestrial ecosystems: a dataset derived from a gross primary productivity product based on solar-induced chlorophyll fluorescence" (ID: ESSD-2021-452).

Those comments were very helpful for revising and improving our manuscript. We have studied the comments carefully and have made corrections which we hope meet with approval. A revised manuscript is being submitted for your consideration. Please see below for the point-to-point response. We have also highlighted the changes we have made in blue text (not including changes to the references) in our main manuscript (i.e. the track-changes file).

We are looking forward to your further decision.

Yours Sincerely,

Jing Fang, on behalf of all authors

**Reviewer 1**

Fang et al. generated a 0.05 degree, annual dataset of vegetation photosynthetic phenology metrics (i.e., SOS, EOS, LOS) in North Hemisphere (NH) terrestrial ecosystems during 2001-2020. There are two major innovations in this study. First, a solar-induced chlorophyll fluorescence (SIF) derived GPP (GOSIF-GPP) product was utilized to derive phenology metrics, because SIF has been demonstrated to be a better proxy for photosynthesis than other vegetation indices (VIs). Second, a method that combined smoothing splines with multiple change-point detection was employed, so that phenology metrics, especially with multiple growing seasons, could be derived accurately. Fang et al. found SIF derived GPP could provide a better phenology estimation than other VIs, when validating with the metrics inferred from flux tower GPP. Further, the new dataset showed a trend of advanced SOS, delayed EOS, and extended LOS over 60-70% of the land area in NH. I believe this dataset will be useful for future studies to examine vegetation phenology under climate change and benchmark Earth System Models.

The data is well archived and the analysis is generally solid. However, this manuscript still requires some revision work, especially to clarify some ambiguous expressions so that readers will not be confused about. Please see my comments below.

**RESPONSE: We thank the reviewer for checking our manuscript carefully, and we think the reviewer's comments are very important to our work. We checked and clarified the ambiguous expressions in the whole of our manuscript. The responses to the reviewer's concerns have been included in each comment.**

Major comments:

1. In Section 4.1, the authors utilized R and RMSE to evaluate the phenology metrics estimated from different datasets. I suggest the authors checking the mean bias as well. In addition, I suggest the authors making a comparison separately for each land cover

to evaluate the performance of each biophysical variable.

**RESPONSE: Yes, we also considered that mean bias was a good statistic for the phenology metrics. We added the bias in the new Table 1 (see the following table: Table 1). We found the best performance of bias in SOS was the GOSIF-GPP, while in EOS and LOS was the BESS-GPP. In general, GOSIF-GPP showed the best performance of all statistics (i.e. R, RMSE, and Bias) in all phenology metrics. We added the calculation method of bias in Eqn 6 and the corresponding description of bias results in line 273-276 (blue text).**

**For the different land covers, we added a table to compare the performance (see the following table: Table S2). The new Table S2 showed the performance of the GOSIF-GPP phenology in the different terrestrial ecosystems. The GOSIF-GPP phenology had a high performance for grasslands, wetlands, and croplands (R>0.84), and a moderate performance for forests and shrublands (R=0.77 and 0.51, respectively). However, the results were limited by the number of observation sites in different ecosystems (e.g. forests had 295 site-year data, while wetlands had only 10 site-year data). Therefore, more tower sites need to be considered in further studies so that the photosynthesis phenology metrics from the SIF product can be better evaluated. We added the above description in line 303-310 (blue text).**

**Table 1.** Statistical comparison of the phenology metrics retrieved from EC tower GPP and GOSIF-GPP, NDVI, EVI, NIRv, MODIS-GPP, GLASS-GPP, and BESS-GPP. 10%, 25%, and 50% mean the amplitude thresholds. The bold means the highest *R* and the lowest *RMSE*. *R*: correlation coefficient; *RMSE*: root mean square error; *Bias*: mean bias.

| Data source | SOS 10% | SOS 25% | SOS 50% | EOS 10% | EOS 25% | EOS 50% | LOS 10% | LOS 25% | LOS 50% |
|---|---|---|---|---|---|---|---|---|---|
| | | | | | *R* | | | | |
| GOSIF-GPP | **0.79** | **0.80** | **0.78** | **0.63** | **0.73** | 0.63 | **0.72** | **0.76** | **0.65** |
| NDVI | 0.14 | 0.25 | 0.39 | 0.45 | 0.42 | 0.56 | 0.28 | 0.32 | 0.40 |
| EVI | 0.40 | 0.46 | 0.57 | 0.57 | 0.60 | 0.66 | 0.37 | 0.37 | 0.38 |
| NIRv | 0.47 | 0.51 | 0.60 | **0.63** | 0.66 | 0.67 | 0.51 | 0.48 | 0.41 |
| MODIS-GPP | 0.66 | 0.67 | 0.65 | 0.29 | 0.55 | 0.61 | 0.47 | 0.55 | 0.49 |

| | | | | | | | | | |
|---|---|---|---|---|---|---|---|---|---|
| GLASS-GPP | 0.65 | 0.67 | 0.67 | 0.28 | 0.59 | 0.63 | 0.46 | 0.53 | 0.48 |
| BESS-GPP | 0.72 | 0.74 | 0.75 | 0.41 | 0.60 | **0.70** | 0.56 | 0.60 | 0.59 |
| ***RMSE* (days)** | | | | | | | | | |
| GOSIF-GPP | **18.03** | **15.83** | **14.99** | **23.55** | 21.89 | 24.38 | **33.93** | **29.14** | 27.89 |
| NDVI | 36.91 | 32.18 | 26.20 | 34.13 | 39.68 | 41.86 | 53.87 | 53.92 | 52.39 |
| EVI | 36.97 | 31.34 | 24.97 | 31.75 | 29.41 | 26.09 | 58.53 | 49.04 | 39.28 |
| NIRv | 29.86 | 26.00 | 21.74 | 27.26 | 25.67 | 24.95 | 46.11 | 40.03 | 35.12 |
| MODIS-GPP | 22.98 | 20.56 | 18.83 | 30.22 | 24.76 | 23.88 | 43.79 | 36.33 | 32.25 |
| GLASS-GPP | 25.15 | 23.47 | 22.17 | 29.09 | 23.8 | 24.07 | 46.88 | 41.11 | 39.41 |
| BESS-GPP | 20.04 | 17.43 | 15.24 | 25.7 | **21.59** | **19.16** | 38.23 | 32.21 | **26.61** |
| ***Bias* (days)** | | | | | | | | | |
| GOSIF-GPP | -3.73 | **-2.72** | **-1.31** | 9.05 | 10.55 | 10.05 | 12.78 | 13.26 | 11.36 |
| NDVI | **-0.98** | 4.08 | 4.27 | -10.94 | -19.96 | -26.9 | -9.9 | -24.0 | -31.2 |
| EVI | -17.84 | -12.04 | -8.33 | 11.89 | 5.27 | 0.77 | 29.74 | 17.31 | 9.10 |
| NIRv | -13.68 | -9.53 | -6.98 | 9.17 | 3.67 | **0.20** | 22.85 | 13.19 | 7.18 |
| MODIS-GPP | 6.37 | 5.47 | 4.0 | 1.60 | 2.30 | 3.27 | -4.77 | **-3.18** | **-0.73** |
| GLASS-GPP | 12.67 | 13.47 | 13.53 | -7.29 | -7.51 | -9.54 | -19.9 | -21.0 | -23.0 |
| BESS-GPP | -4.02 | -3.39 | -2.43 | **0.88** | **1.03** | 0.94 | **4.90** | 4.42 | 3.37 |

**Table S2.** Statistical comparison of the phenology metrics retrieved from EC tower GPP and GOSIF-GPP in the different terrestrial ecosystems. 10%, 25%, and 50% mean the amplitude thresholds. N: number of the site-years; *R*: correlation coefficient; *RMSE*: root mean square error; *Bias*: mean bias.

| | Vegetation types | SOS 10% | SOS 25% | SOS 50% | EOS 10% | EOS 25% | EOS 50% | LOS 10% | LOS 25% | LOS 50% |
|---|---|---|---|---|---|---|---|---|---|---|
| **R** | Forests (N=295) | 0.77 | 0.77 | 0.70 | 0.44 | 0.31 | 0.09 | 0.72 | 0.69 | 0.56 |
| | Shrublands (N=11) | 0.51 | 0.69 | 0.69 | 0.73 | 0.76 | 0.34 | 0.98 | 1.00 | 0.80 |
| | Savannas (N=13) | 0.28 | 0.85 | 0.92 | 0.28 | 0.20 | 0.22 | 0.39 | 0.54 | 0.74 |
| | Grasslands (N=13) | 0.95 | 0.95 | 0.95 | 0.84 | 0.81 | 0.05 | 0.93 | 0.93 | 0.64 |
| | Wetlands (N=10) | 0.95 | 0.99 | 0.99 | 0.46 | 0.30 | 0.38 | 0.99 | 0.91 | 0.87 |
| | Croplands (N=47) | 0.84 | 0.77 | 0.72 | 0.49 | 0.54 | 0.60 | 0.31 | 0.42 | 0.26 |
| **RMSE (days)** | Forests (N=295) | 17.84 | 15.51 | 14.84 | 21.38 | 21.29 | 24.09 | 33.02 | 29.99 | 28.23 |
| | Shrublands (N=11) | 31.68 | 30.60 | 30.06 | 36.89 | 28.51 | 18.39 | 19.35 | 13.95 | 4.20 |
| | Savannas (N=13) | 23.62 | 22.86 | 19.10 | 32.32 | 31.70 | 18.98 | 31.32 | 22.77 | 12.92 |
| | Grasslands (N=13) | 14.29 | 13.07 | 11.83 | 12.34 | 11.51 | 36.01 | 22.98 | 20.31 | 39.90 |
| | Wetlands (N=10) | 12.77 | 10.09 | 11.30 | 10.28 | 10.49 | 10.53 | 12.69 | 16.90 | 17.55 |
| | Croplands (N=47) | 22.92 | 20.65 | 19.41 | 48.99 | 33.21 | 25.04 | 58.41 | 32.17 | 20.68 |
| **Bias (days)** | Forests (N=295) | -6.14 | -4.87 | -2.86 | 12.44 | 12.15 | 12.30 | 18.59 | 17.01 | 15.16 |
| | Shrublands (N=11) | -15.27 | -14.27 | -8.64 | 29.09 | 20.82 | 8.82 | 4.33 | 2.67 | -0.33 |
| | Savannas (N=13) | 21.10 | 22.30 | 18.70 | 3.50 | 21.70 | 12.80 | -17.60 | -0.60 | -5.90 |
| | Grasslands (N=13) | -7.94 | -5.50 | -4.44 | 2.94 | 1.19 | -12.13 | 10.88 | 6.69 | -7.69 |
| | Wetlands (N=10) | 9.13 | 7.50 | 8.63 | -1.38 | -1.00 | -0.13 | -10.50 | -8.50 | -8.75 |
| | Croplands (N=47) | 15.71 | 12.65 | 8.88 | -31.65 | -7.24 | -0.47 | -47.35 | -19.88 | -9.35 |

2. For figures, the North Pole with no vegetation was in the center, while the vegetated continents were kind of clumped and hard to recognize. Especially, it is not

straightforward to find California and North China Plain mentioned in the text. I suggest the authors using a different projection to present the results.

**RESPONSE: This was a helpful suggestion for presenting our results. The normal projection would showed indistinguishable in some figures (e.g. Fig. 4, 3×3 subplot) because the paper size of ESSD was 21×24 cm. Limited by the paper size, we used the projection of Equidistant Azimuthal (i.e. circular projection) in this manuscript. To make the reviewer and the readers to better distinguish the locations of geography, we provided a comparison between the different projections as a reference in the supplementary materials. The comparison was between the projection of Equidistant Azimuthal (i.e. circular projection) and the projection of Bolshoi Sovietskii Atlas Mira (i.e. rectangular projection). This comparison had shown in Fig. S1 (see the following figure: Fig. S1). We also added the necessary lines and names of latitude in Fig. 3 and Fig. S1 (see the following figure: Fig. 3 and Fig. S1).**

[Figure]

**Fig. S1. The spatial distribution of vegetation types and EC tower sites under the different projections in the Northern Hemisphere (0.05º spatial resolution). The left part uses the projection of Equidistant Azimuthal, and the right part uses the projection of Bolshoi Sovietskii Atlas Mira.**

[Figure]

[Figure]

Single     Double

Fig. 3. The spatial distribution of the number of growing seasons in the Northern Hemisphere (0.05º spatial resolution). The double seasons mean there are two photosynthesis cycles in one year. We used the Pruned Exact Linear Time (PELT) method to detect the change points of each photosynthesis cycle.

Some minor suggestions:

Line 13 Specify which vegetation indices were used for comparison in this study.

**RESPONSE: Here, we added the specific name of the indices, in line 13 (blue text).**

Line 54 Clarify "better" compared to what, or just use "well".

**RESPONSE: We had revised the word to 'well', in line 54 (blue text).**

Line 56 "instantaneous" refers to changes happening in a short time (e.g., diurnal), therefore may be not accurate to be used here. The point should be: these traditional VIs can mainly detect structural changes but are less sensitive to physiological changes.

**RESPONSE: According to the reviewer's comment, we changed the sentence to 'these indices work well for capturing the variations in chlorophyll content or structural changes but are less sensitive to physiological changes in vegetation photosynthesis', in line 55-56 (blue text).**

Line 64 Clarify "more accurately" compared to what.

**RESPONSE: Here, we deleted the word 'more'.**

Line 72 Explain a bit more about "predetermined thresholds or inflection points".

**RESPONSE: We had added an example to explain this sentence: 'predetermined thresholds or inflection points (e.g. using the peaks in the second derivative as the points)', in line 71-72 (blue text).**

Line 74 Explain what kind of "reconstruct" is referred to here, smoothing?

**RESPONSE: The 'reconstruct' in this sentence meant some previous studies used the double-sigmoidal logistic model to acquire the data curve. We added a sentence to describe this: 'reconstruct the original data sequence by using double-sigmoidal logistic model', in line 74-75 (blue text).**

Line 78 "of" -> "which combined"

**RESPONSE: We had revised this word, in line 79 (blue text).**

Line 80 "The strength of this method is not limited by… and can also be applied…" -> "This method has great strength in two aspects: (1) it is not limited by…; (2) it can also be applied…"

**RESPONSE: We had rewritten this sentence, in line 80-82 (blue text).**

Line 83 "needs to be extended" -> "can be further extended"

**RESPONSE: We had changed this sentence, in line 84 (blue text).**

Line 89 "constructed" -> "adopted"/"developed"

**RESPONSE: We revised the word to 'developed', in line 90 (blue text).**

Line 94 "SIF-GPP" -> "GOSIF-GPP"

**RESPONSE: We had revised this word, in line 95 (blue text).**

Line 110 Clarify which classification type was used, IGBP?

**RESPONSE: Yes, you were right. We used the IGBP classification. We had added the classification type in line 110 (blue text).**

Line 117 Clarify at which time scales the maximum GPP was calculated, e.g., 8-day maximum over 2001-2020?

**RESPONSE: We had corrected the description to '8-day maximum GPP over 2001-2020', in line 119.**

Line 122 Clarify which type of partitioning approach was used for FLUXNET 2015 GPP?

**RESPONSE: We clarified that we used GPP based on the Variable Ustar Threshold (VUT) mean values from the FLUXNET2015 Dataset, in line 125 (blue**

**text).**

Line 125 Clarify how the sites were determined as "relatively homogeneous"?

**RESPONSE: In this study, one given site was considered homogeneous when the dominant land cover type was similar to that of the site. We clarified this 'relatively homogeneous' in line 129-131 (blue text).**

Line 136 Briefly describe how the MODIS GPP dataset was derived.

**RESPONSE: The MODIS GPP data was extracted from the MOD17A2H dataset and it was generated by Moderate Resolution Imaging Spectroradiometer (MODIS) Leaf Area Index (LAI)/Fraction of Photosynthetically Active Radiation (FPAR). We describe the dataset in line 140-143 (blue text).**

Line 136 Clarify why the time period "from 2001 to 2014" was selected for comparison?

**RESPONSE: This period was the observed time of EC towers. To avoid make misunderstanding for the reviewer and readers, we used the 'The time period of all data was consistent with the observations of EC towers.' instead of 'from 2001 to 2014', in line 140-143 (blue text).**

Line 152-153 I suggest separately describing why (1) smoothing splines and (2) change points were used in this study. It seems the smoothing splines is to minimize the influence of outliers (as mentioned later in the paragraph) and not specifically aim to resolve multiple cycles.

**RESPONSE: Thanks a lot for your suggestion, we revised this sentence to: 'we used (1) the smoothing splines to minimize the influence of outliers; (2) the change points to identify the transition dates of photosynthesis.', in line 159-161.**

Line 163 Clarify what "change characteristics" refers to, seasonal cycle?

**RESPONSE: The 'change characteristics' referred to 'reduce the influence of outliers and retain the major change characteristics of the original data sequence'.**

**We clarified this in line 170 (blue text).**

Line 166 How about the case when ratio is larger than one standard deviation ABOVE the mean ratio?

**RESPONSE: We missed the description for the case above the mean ratio. Indeed, we used the same method to handle the below and the above cases. We revised the words to 'below or above', in line 174 (blue text).**

Line 169 "by reconstructing the data time series by estimating parameters in the double logistic model" -> "if reconstructing the data time series with a double logistic model"

**RESPONSE: We had changed this sentence in line 177 (blue text).**

Line 176 Briefly explain what the "penalty factor" was used for.

**RESPONSE: We explained the 'penalty factor' as: 'The penalty factor was acted to limit the number of returned significant changes by applying the additional penalty to each prospective changepoint.', in line 185-187 (blue text).**

Line 182 Explain what the "baselines" were used for.

**RESPONSE: Here, the difference between the baselines was used as the amplitude (i.e. the amplitude was equal to the peak baseline minus the bottom baseline.). We explain it in line 192-193 (blue text).**

Line 185-186 Explain what "amplitude thresholds" mean.

**RESPONSE: The 'amplitude thresholds' meant the value reached 10%, 25%, and 50% of the amplitude. The amplitude was equal to the peak baseline minus the bottom baseline. We explain this in line 197 (blue text).**

Line 188 I am not sure what "the most tightly-constrained transition dates" means.

**RESPONSE: This meant 'the accurate transition dates.'. We explained the sentence in line 200 (blue text).**

Eq1, 2. Does this assume Bottom1 and Bottom2 are roughly zero? Fig 1 showed above-zero values. How would this affect your threshold calculation?

**RESPONSE: Bottom1 and Bottom2 were not equal to zero. According to the time series of mean GPP value, we used the difference method to detect the bottoms and peaks (i.e., minimum and maximum value in each cycle). The Bottom1 and Bottom2 were equal to the value of these bottoms (see line 188-190).**

Line 195 "smoothing splines" -> "smoothed splines"?

**RESPONSE: We had revised this word.**

Line 210-211 The current expression about uncertainty quantification is ambiguous. I am not sure if I fully understand. Could you elaborate more on this?

**RESPONSE: Here, we added the explanation to describe: 'For each year of the individual grid cell, we used bootstrapping to replace the transition dates with 100 times random uniform sampling (Yang and Noormets 2021). Bootstrapping was a statistical procedure that resampled a single dataset to create many simulated samples. In this study, each grid had 100 transition dates created by the bootstrapping method. The 5th and 95th percentiles of the 100 bootstrapped data were considered as the confidence interval of the mean estimated from the original transition dates.', in 222-228.**

Line 229 Was the correlation coefficient calculated across all the site-years?

**RESPONSE: Yes, the correlation coefficient was calculated across all the site-years. We added this description in line 244 (blue text).**

Line 255 "that" -> "which showed that". And "uncertainty occurred" is not accurate, as phenology estimation from GOSIF-GPP also present uncertainties (Section 4.4).

**RESPONSE: To make this sentence clearer, we revised it to: 'this was consistent**

**with previous studies which showed that the larger uncertainty occurred in satellite-based EOS estimations than the SOS estimations', in line 271-272 (blue text).**

Line 262 "the method" -> "the proposed method"

**RESPONSE: We had added the word 'proposed' in line 279 (blue text).**

Line 264-265 "part of the cropland" -> "some croplands"

**RESPONSE: We had revised these words in line 281 (blue text).**

Line 284-286 I am not sure what information the authors would like to convey here.

**RESPONSE: We deleted this sentence to keep the concise description.**

Line 296 "10% SOS" -> "SOS10%" to be consistent with Line 223

**RESPONSE: We had changed the expression to 'SOS$_{10\%}$', in line 311 (blue text).**

Line 299 "two different mixed grids" -> "two different kinds of mixed grids"

**RESPONSE: We had revised this sentence in line 314 (blue text).**

Line 300 "another" -> "the other"

**RESPONSE: We had revised this word in line 315 (blue text).**

Line 338 The analysis seems still long-term trend, not "interannual variation".

**RESPONSE: We thought this expression may be inappropriate. We revised the 'interannual variation' to the 'trends', in line 353 (blue text).**

Line 359 How did you define "re-modeling of the GPP time series"? Is the smoothing procedure employed in this study not a kind of "re-modeling"?

**RESPONSE: Here, the 're-modeling' method was used in some studies and it meant using the double logistic model and the corresponding model parameters to**

**simulate the GPP time series. In this study, we used the filter method to smooth the GPP time series instead of 're-modeling'. We added the explanation to describe the 're-modeling', in line 374-375 (blue text).**

Line 362 "most appropriate for their specific application" is ambiguous and confusing.

**RESPONSE: We had added an example to explain it: 'e.g. the users can choose the 10% (or 25% and 50%) threshold to acquire the earlier (or later) transition dates during the photosynthesis rising (or falling) stage.', in line 378-379 (blue text).**

Line 368-369 Describe the "spatially explicit pattern".

**RESPONSE: We described it as: 'the SOS was delayed, the EOS was advanced, and the LOS was extended gradually with the latitude increased', in line 386-388 (blue text).**

Fig 2 Figure labels are confusing. X and y variables should be evaluated at the same threshold. Only adding the threshold to y axis label is misleading. "GOSIF-GPP", "NDVI", etc., in the top of the figure are also confusing, shouldn't they be added to the y axis labels?

**RESPONSE: According to the reviewer's comment, we transposed the X and Y axis in the new Fig. 2 (see the following figure: Fig. 2). We also added 'GOSIF-GPP', 'NDVI', etc., to the Y-axis label.**

[Figure]

**Fig. 2. The comparison of the phenology metrics retrieves from EC tower GPP (EC-GPP) and GOSIF-GPP, NDVI, EVI, NIR$_V$, MODIS-GPP, GLASS-GPP, and BESS-GPP. Each subplot has 389 site-year data. The significant correlations of all results are less than 0.05 ($p<0.05$). The solid line represents a 1:1 line. SOS: start time of the growing season; EOS: end time of the growing season; LOS: length of the growing season; DOY: day of the year; R: correlation coefficient.**

Two suggestions for archiving the data:

1. The authors can set the data type as integer to reduce the file size.

**RESPONSE: This was a helpful suggestion. We had set the data type as integer and reduced the file size (the new dataset had uploaded in the 'figshare': https://doi.org/10.6084/m9.figshare.17195009.v3).**

2. Set NA values for the ocean.

**RESPONSE: We had set NA values for the ocean in the new dataset (see: https://doi.org/10.6084/m9.figshare.17195009.v3).**

---

## Author Comment (AC2)

**RESPONSE TO REVIEWERS' COMMENTS**

Dear Prof. Tian and reviewer,

We would like to express our great appreciation to you for handling and reviewing our manuscript entitled "Vegetation photosynthetic phenology metrics in northern terrestrial ecosystems: a dataset derived from a gross primary productivity product based on solar-induced chlorophyll fluorescence" (ID: ESSD-2021-452).

Those comments were very helpful for revising and improving our manuscript. We have studied the comments carefully and have made corrections which we hope meet with approval. A revised manuscript is being submitted for your consideration. Please see below for the point-to-point response. We have also highlighted the changes we have made in blue text (not including changes to the references) in our main manuscript (i.e. the track-changes file).

We are looking forward to your further decision.

**Yours Sincerely**,

**Jing Fang, on behalf of all authors**

**Reviewer 2**

This study developed a photosynthetic phenology metric dataset from 2001 to 2020 with SIF-based GPP and the retrieval of phenology. This has important implications for the modeling and analysis of the global carbon cycle. However, I believe the comparison and validation approach proposed is flawed in this manuscript, making a reliable assessment challenging. Consequently, the current manuscript is not suitable for publication in the ESSD journal.

**RESPONSE:** We thank the reviewer for the helpful comments. For the main concern of the reviewer, we remade the phenology comparison and validation from more GPP data sources in the manuscript. Detailed explanations of the concerns can be found in the following items as responses to each concern.

The specific suggestions are as follows:

**Main comments**

(1) the comparison for phenology metrics: The vegetation greenness and photosynthesis are not always coupled (this is mentioned in the Introduction section). This study conducted a comparison between GPP-based and VI-based phenology metrics to prove that the GPP-based metrics outperform. In my opinion, this is not directly comparable. In contrast, GPP-based phenology metrics are based on vegetation photosynthesis activity, whereas VI-based phenology metrics (NDVI, EVI) are based on vegetation morphology, structure, and greenness. NDVI/EVI (greenness index) cannot well account for most productivity variation than GPP products. In addition, the remotesensing VIs are derived from observation while GPP is derived from simulation. So, I suggest authors can replace VIs with multiply GPP products (excepting MODIS-GPP in this manuscript), and further comparing with SIF-based GPP.

**RESPONSE:** Yes, we agreed that the phenology metrics of vegetation indices (i.e. VI) were based on structure and greenness, and the phenology metrics of GPP

were based on the photosynthesis activity. The comparison from one GPP product was not enough. Adding multiply GPP products to compare was a helpful suggestion for our study. In the new manuscript, we added two additional GPP **GLASS-GPP** product products: (download from: http://www.glass.umd.edu/Download.html) and BESS-GPP product (download from: https://www.environment.snu.ac.kr/bess-flux). We extracted the phenology metrics of these data and we had a total of four GPP products for comparison (i.e. GOSIF-GPP, MODIS-GPP, GLASS-GPP, and BESS-GPP). Here, we introduced GLASS-GPP and BESS-GPP briefly: 'The GLASS-GPP data was generated by a light use efficiency (EC-LUE) model and the environmental variables (i.e. atmospheric CO2 concentration, radiation components, and atmospheric vapor pressure deficit) (Zheng et al. 2020). The BESS-GPP data was generated by a simplified process-based model, the Breathing Earth System Simulator (BESS), and MODIS Atmosphere and Land products (Jiang et al. 2016).', in line 145-150 (blue text). The results of phenology metrics extracted from the additional GPP products could be found in the next response.

(2) validation: The derivative datasets from EC-GPP were used for the validation. The derivative datasets from EC-GPP fall into the category of photosynthetic phenology. Hence, a tendentious validation generates a bias toward phenology metrics in the two categories. This verification is more suitable for photosynthetic phenology than for structure. The results that the accuracy of GPP-based phenology metrics outperforms the VI-based ones are not solid. Suggest authors validate photosynthetic phenology results using photosynthetic phenology observation.

**RESPONSE:** We also thought the EC-GPP was fallen into the category of photosynthetic phenology. As above mentioned, we added the phenology metrics from GLASS-GPP and BESS-GPP. We focused on the comparison between the different GPP products, and the VI products were only as the reference. In the revised manuscript, we remade Fig. 2 to present the comparison and validation of the multiple GPP products (see the following figure: Fig.2). Overall, the phenology

metrics of GOSIF-GPP showed the highest correlations with the phenology metrics of EC-GPP. The BESS-GPP performs slightly worse than the GOSIF-GPP. The MODIS-GPP and GLASS-GPP had larger deviations compared to GOSIF-GPP. We added the description of these results in line 283-285 (blue text).